# Transposon-directed insertion-site sequencing (TraDIS) analysis of *Enterococcus faecium* using nanopore sequencing and a WebAssembly analysis platform

Alexandra L. Krause,[1] Wytamma Wirth,[1,2] Adrianna M. Turner,[1] Louise Judd,[1,2] Lucy Li,[1] Willem van Schaik,[3] Benjamin P. Howden,[1,2,4] Glen P. Carter,[1] Torsten Seemann,[1,2] Ryan Wick,[1,2] Timothy P. Stinear,[1,2] Andrew H. Buultjens,[1,2] Ian R. Monk[1]

**ABSTRACT** Vancomycin-resistant *Enterococcus faecium* (VREfm) are healthcare-associated opportunistic pathogens of global significance. Genetic tools are needed to understand the molecular basis for VREfm clinically relevant phenotypes, such as persistence within the human gut or antimicrobial resistance. Here, we present a transposon-directed insertion-site sequencing (TraDIS) platform optimized for *E. faecium*. We engineered a transposon delivery plasmid, pIMTA(*tetM*), that can generate high-density transposon mutant libraries, combined with Oxford Nanopore Technology amplicon sequencing to map the transposon insertion sites. We have also customized a bioinformatic analysis suite that includes a WebAssembly powered visualization tool called *Diana*, for TraDIS data exploration and analysis (https://diana.cpg.org.au/). To demonstrate the performance of our platform, we assessed the impact of vancomycin exposure on a library of 48,458 unique transposon mutants. As expected, we could confirm the importance of the *vanB* operon for VREfm vancomycin resistance. However, we also identified an essential role for both *vanWB* and *vanYB,* each previously designated as protein of unknown function and accessory for resistance, respectively. Our end-to-end platform for running TraDIS experiments in VREfm will permit accessible, genome-scale, forward genetic screens to probe molecular mechanisms of persistence and pathogenesis.

**IMPORTANCE** There are limited genetic tools specifically developed and optimized for function in *Enterococcus faecium*. Here, we addressed this gap through the development of a transposon-directed insertion-site sequencing platform with a plasmid we engineered to specifically function in *E. faecium*. The application of nanopore sequencing, with a highly accessible sequence data processing and bioinformatic analysis pipeline, streamlines and simplifies the methodology. These developments will allow the functional genomic analysis of important traits involved in the pathobiology of this understudied bacterium. The approach and tools we have described here are likely applicable to other Gram-positive bacteria.

**KEYWORDS** transposons, DNA sequencing, *Enterococcus faecium*, TraDIS, fitness, vancomycin resistance

Vancomycin-resistant *Enterococcus faecium* (VREfm) is an opportunistic human pathogen and a major cause of healthcare-associated infections (1, 2). A combination of extensive antibiotic resistance and poor transformation efficiency of VREfm clinical isolates has made the application of molecular methods to analyze phenotypes for persistence and pathogenesis challenging (3). To facilitate genotype-to-phenotype analyses, we have established a transposon-directed insertion-site sequencing (TraDIS)

**Peer Reviewer** Emma Rachel Holden, Quadram Institute, Norwich, Norfolk, United Kingdom

Address correspondence to Ian R. Monk, ian.monk@unimelb.edu.au.

The authors declare no conflict of interest.

See the funding table on p. 13.

platform to enable future applications such as dissecting the molecular basis for the environmental hardiness of this bacterium. TraDIS is a powerful tool for linking bacterial phenotypes with genotypes and to map gene interaction networks (4–6). In TraDIS, a bacterial population of random transposon mutants is challenged with specific conditions and then quantified through sequencing of transposon insertion sites, compared to an unchallenged library. Analysis of transposon enrichment or loss across a genome can identify genes involved in survival/growth under the challenge condition (4, 6–8). The ideal TraDIS library would thus comprise a population of bacterial cells that each had a single transposon insertion at each nucleotide position in the genome. For various reasons, including transposon insertion specificity, insertion biases, and inadequate transposase induction, this is never achieved (9–11). In low-GC Gram-positive bacteria, the mariner *himar1* transposon system exhibits excellent characteristics for TraDIS due to the sole requirement of a TA dinucleotide motif for transposon insertion via a cut-and-paste mechanism (12–14). For the quantitative analysis of TraDIS transposon insertion data, numerous tools have been established, including a Galaxy-hosted workflow for the in-depth examination of transposon library insertions. This workflow has been used to process TraDIS data from libraries sequenced on the Illumina platform and uses two different bioinformatic tools that apply multiple statistical analyses (Bio-Tradis and TRANSIT) to determine conditionally and constitutively essential genes (15–17). TRANSIT is a *Python*-based software tool designed for the analysis of *himar1* Tn-Seq data, offering a graphical interface to three statistical methods for identifying essential genes and conducting comparative analyses across different conditions (16).

There are four main steps in the TraDIS protocol: (i) transposon library construction, (ii) library challenge, (iii) transposon insertion site recovery, and (iv) sequencing and analysis (Fig. S1). There has been one report of a successful transposon sequencing (Tn-Seq) library for *E. faecium* (18). The plasmid pGPA1 was constructed to deliver the *himar1* transposon into the VanA clinical *E. faecium* strain E745 (18) and has subsequently been deployed in strains *E. faecium* E980 (19) and *E. faecium* E8202 (20). We were unsuccessful in the construction of complex libraries with pGPA1 using a different clinical isolate, despite repeated attempts. The reason(s) for this failure were not apparent. In the present study, we constructed a new transposon delivery plasmid called pIMTA(*tetM*), which we paired with TraDIS for genomic library construction and applied nanopore amplicon rather than Illumina sequencing to identify the transposon insertion sites. The processed reads were then visualised and analyzed using Bio-Tradis integrated into the "Diana" online tool, which streamlines the analysis of transposon insertions across a genome and between different experimental conditions (17).

## MATERIALS AND METHODS

### Bacteria, plasmids, and growth conditions

The *E. faecium* ST796 VanB strain AUS0233Δ*tetM* (described below) used in this study was grown at 37°C with shaking at 200 rpm, in Brain Heart Infusion (BHI) broth (Bacto, Difco). The addition of antibiotics and other compounds to media was added at the following concentrations: chloramphenicol (Cm) 10 µg/mL, glycine (3% wt/vol), vancomycin 4 µg/mL, tetracycline (Tet) 10 µg/mL, and 500 ng/mL anhydrotetracycline (aTc). Bacterial growth was determined at $OD_{600}$ using a Biophotometer (Eppendorf). For PCR, Phusion high-fidelity DNA polymerase and Phire Green Hot Start II DNA Polymerase (Thermo Fisher) were used following the manufacturer's instructions. All restriction endonuclease enzymes used were purchased from New England Biolabs (NEB).

### Construction of the TraDIS plasmid pIMTA(tetM)

The pIMTA(*tetM*) vector was constructed as follows. The broad-host range pWV01(ts) replicon was amplified from pGPA1 with primers IM1377/IM1440 (Table 1), as described previously (18, 21). The chloramphenicol resistance gene was PCR amplified and

**TABLE 1** Oligonucleotides used in this study[a]

| Primer | Sequence (5′–3′)[a] | Target | Template | Source |
|---|---|---|---|---|
| IM1377 | ATATAGATCTGACTCCCGTTGATAGATCCAGTAATGACCTCAGAACTCCATCT GGATTTGTTCAGAAGCGTCGGTTGCCGCCGGGCGTTTTTATTGGTGAGAATGGTACCATATATGAGCTC CCAAGCACTAGGCGATTTTTATTAAAACG | pWV01(ts) rep | pGPA1 | (18) |
| IM1440 | TCGAGTGAGGGTGTCAAAATTCCTTATTTATTTCCCCGTTTCAGCATC | pWV01(ts) rep | pGPA1 | (18) |
| IM1439 | AGGAATTTTGACACCCTCACTCGAATGTGCTATAATGGCCACAAAGGAGGA AGGATCAATGAACTTTAATAAAAATTGATTTAGACAATTG | Pha-1-cat | pIMAY-Z | (22) |
| IM1380 | ATATAGATCTATATGCATGCCAAATAAAACGAAAGGCTCAGTCGAAAGACT GGGCCTTTCGTTTTATCTGTTGTTGTCGGTGAACGCTCTCCTGAGTAGGAC AAATCTGCAGATATATCCTCCTTTATAAAAGCCAGTCATTAGGCCTATCTG | Pha-1-cat | pIMAY-Z | (22) |
| IM1391 | GCCAACCTGTTAGAATTCCTGCAGCCAGTCAAAAGCCTCCGACCGGA GGCTTTTGACTAAGGTTATGCTGCTTTTAAGACCCAC | tetR-C9 | pIMC9 | Unpublished plasmid |
| IM1392 | ATATAGATCTTTATTCAACATAGTTCCCTTCAAGAGC | tetR-C9 | pIMC9 | Unpublished plasmid |
| IM1389 | ATATGCATGCTAACAGGTTGGCTGATAAGTCC | ermC | pJZ037 | (25) |
| IM1611 | TTTCCATAACTTTAGCTGCAGGTACC | tetM | AUS0233 | This study |
| IM1612 | TTCTATGAGTGCGCTTTGTAAATTGG | tetM | AUS0233 | This study |
| IM1613 | GGTACCTGCAGCTAAAGTTATGGAAACAAATATTGGTACATTATTACAGCTATTTG | tetM | AUS0233 | This study |
| IM1614 | AATTTACAAAAGCGACTCATAGAACTAAGTTATTTTATTGAACATATCGTACTTTATC | tetM | AUS0233 | This study |
| IM1324 | CCTCACTAAAGGGAACAAAAGCTGGGTACCAGAACTGGTAAATCCTATTCACAATCG | tetM | AUS0233 | This study |
| IM1325 | CATTCAAAAGCCCAAAAGGGCATAAAAATCC | tetM | AUS0233 | This study |
| IM1326 | TATGCCCTTTTGGGCTTTTGAATGTTAGTGTATTTATGTGTTGTTATATAAAATATGGTTTC | tetM | AUS0233 | This study |
| IM1327 | CGACTCACTATAGGGCGAATTGGAGCTCACCTTGTATCGTTA CTTCATGTTTCC | tetM | AUS0233 | This study |
| IM1756 | ACTTGCCTGTGCCTCTATCTTCACCATGTTACTACCGGTGAA CCTGTTTGCC | TraDIS amplicon | TraDIS library | This study |
| IM1757 | ACTTGCCTGTGCCTCTATCTTCAGGTATTCTTAAACTGGGTAC AAAAAACTAAGCCCTCC | TraDIS amplicon | TraDIS library | This study |
| IM1758 | TTTCTGTTGGTGCTGATATTGCTCGGTCTCGGCATTCCTGCT GAACC | Splinkerette primer | TraDIS library | (17) |
| IM1080 | G*AGATCGGTCTGGCATTCCTGCTGAACCGCTCTTCCGATC*T | Splinkerette | - | (17) |
| IM1081 | /5PHOS/G*ATCGGAAGAGCGGTTCAGCAGGTTTTTTTTTC AAAAAAA*A | Splinkerette | - | (17) |

[a]*, phosphorothioate group modification.

included the strong *Bacillus subtilis* promoter *Pha-1* (IM1439/IM1380) from the pIMAY-Z template (22). The replicon and chloramphenicol acetyltransferase (*cat*) marker were joined by SOE-PCR (IM1377/IM1380). The resulting PCR product was digested with BglII, ligated with T4 DNA ligase, and transformed into *E. coli* DC10B-R at 37°C with Cm selection (23). The obtained plasmid was digested with BglII/SphI and ligated to the PCR-amplified *tetR*-C9 transposase (IM1391/IM1392) from pIMC9 (derived from pIMAY with the antisense *secY* replaced with the C9 transposase from pJZ037) (24). Finally, the ITR-flanked *ermC* was PCR amplified from pJZ037 (IM1389) and cloned into the unique SphI site, yielding pIMTA(*ermC*) (25). To replace the *ermC* with *tetM*, the plasmid backbone was PCR amplified (IM1611/IM1612), with the *tetM* gene and native promoter amplified from AUS0233 genomic DNA with (IM1613/IM1614). The two fragments were joined by SLiCE and transformed into DC10B-R, yielding plasmid pIMTA(*tetM*) (Fig. 1).

## Transformation of AUS0233ΔtetM

Electrocompetent *E. faecium* cells were prepared in BHI containing 3% (vol/vol) glycine as described previously (28). Electrocompetent cells stored at −80°C were thawed on ice, 1 µg of pIMTA(*tetM*) added, and incubated on ice for 5 min. The cells were then added to a pre-chilled 0.2 cm electroporation cuvette and pulsed at 2.5 kV, 25 µF, and 200 Ω in a Gene Pulser Xcell electroporator (Bio-Rad). Subsequently, 1 mL of ice-cold BHI broth containing 500 mM sucrose was added, and the cells were recovered at 30°C for 2 h statically before plating onto BHI agar containing 10 µg/mL of Cm and grown for 3 days at 30°C.

## Growth curves of AUS0233 and AUS0233ΔtetM

Strains AUS0233 or AUS0233Δ*tetM* were grown overnight at 37°C at 200 rpm in 10 mL of BHI. The overnight cultures were diluted to a final OD of $_{600}$ 0.0175 in BHI with 200 µL aliquoted in triplicate into a 96-well plate (#3599, Costar). The plate was sealed (optical adhesive cover – Applied Biosystems) and the $OD_{600}$ recorded every 10 min for 10 h with agitation at 300 rpm for 20 s prior to each reading (CLARIOstar$^{plus}$, BMG LABTECH). The data were analyzed with Prism (v10.2.3).

## Targeted deletion of tetM promoter/gene in AUS0233

An allelic exchange product for the deletion of the entire *tetM* gene and promoter region was PCR amplified with primer combinations IM1325/IM1326, IM1327/IM1328, on AUS0233 genomic DNA. The two amplimers were joined by SOE-PCR with IM1325/IM1328, SLiCE cloned into pIMAY-Z and transformed into RbCl-competent *E. coli* DC10B. With the subsequent steps performed as previously described (29). The closed genome sequence of AUS0233Δ*tetM* was hybrid assembled from Illumina and ONT reads using Autocycler v0.1.1, Medaka v2.0.1, Polypolish v0.6.0, and Pypolca v0.3.1 (30).

## TraDIS library preparation

### Induction of transposase

A single colony of AUS0233Δ*tetM* containing pIMTA(*tetM*) from the transformation plate was grown overnight in BHI with 10 µg/mL of Cm at 30°C (shaking at 200 rpm) (Fig. S2). Cells were diluted to $OD_{600}$ 0.01 in 10 mL of BHI broth and grown to $OD_{600}$ ~0.08, transposase induced with 500 ng/mL of aTc and grown for 24 h at 30°C with shaking at 200 rpm. Cells were centrifuged at 5,000 × *g* for 10 min, washed three times in an equal volume of phosphate-buffered saline (PBS), and subsequently resuspended in BHI (Fig. S2).

### Plasmid loss

To stimulate plasmid loss, 1 mL of the washed cells was diluted 1 in 100 in BHI pre-warmed to 42°C and grown overnight at 42°C with shaking at 200 rpm. Serial dilutions of

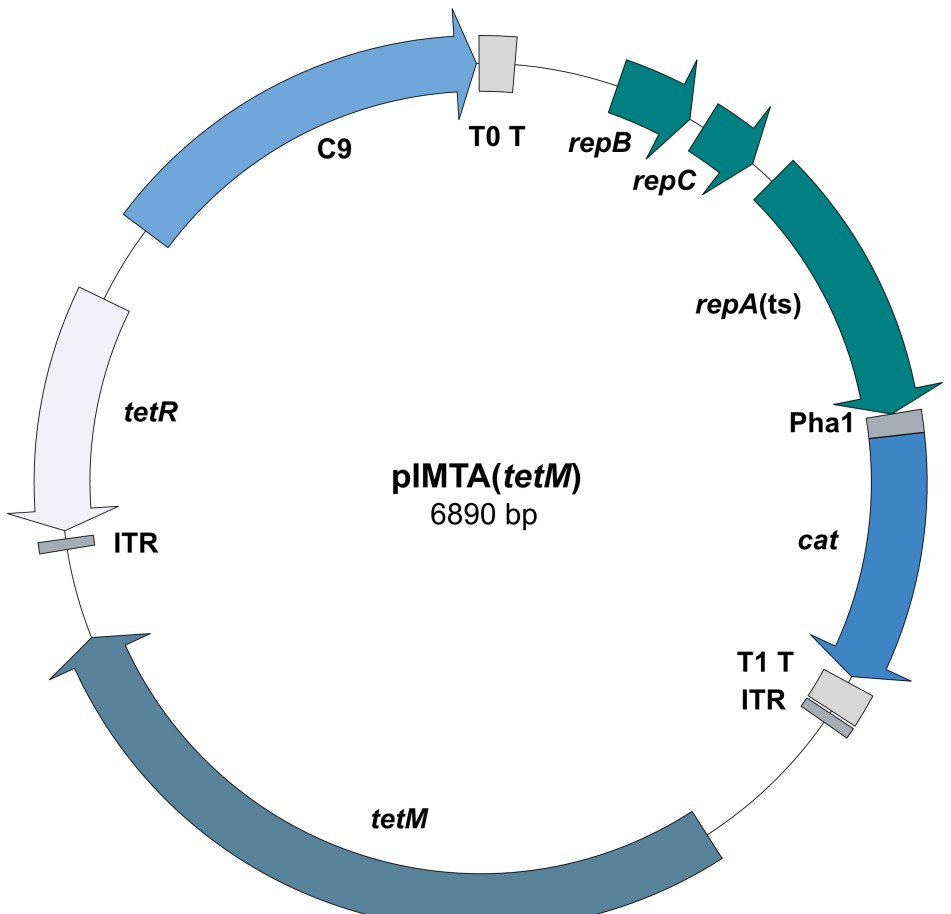

**FIG 1** Map of the transposon mutagenesis plasmid pIMTA(*tetM*). The plasmid pIMTA(*tetM*) contains the complete pWV01 replicon (*repA*(ts)), which is functional at 30°C but not 37°C (21, 26). The C9 transposase (hyperactive himar1 variant) is anhydrotetracycline (aTc) inducible (27). The *tetM* gene, including the native promoter sequence, was introduced between the himar1 inverted terminal repeats (ITR). In the *E. coli* strain DC10B-R, pIMTA(*tetM*) can be propagated at 37°C due to the presence of the wild-type *repA* gene on the chromosome (23). The plasmid also harbours a constitutively expressed cat marker for plasmid selection (22). Transcription terminators, lambda T0 (T0 T) and *rrnB* T1 (T1 T), insulate the replicon from the transposase functions.

the library were plated onto BHI agar (BHIA) containing 10 µg/mL of Tet, and an average concentration of $1.0 \times 10^8$ CFU/mL per library was obtained. Subsequently, 200 colonies were replica-plated onto BHIA containing 10 µg/mL of Tet or Cm (Fig. S2). We routinely obtained >98% plasmid loss using this protocol. After sufficient plasmid loss was confirmed, 25 mL of remaining library culture was combined with 25 mL of 5 M glycerol, and 1 mL aliquots was stored at −80°C until needed.

### Pooling transposon libraries

We attempted to obtain a library density of at least 50,000 unique transposon mutants; however, the frequency of transposition can vary between experiments. Here, six separate transposon libraries were pooled together. Each individual library was plated across 40 BHIA plates containing 10 µg/mL of Tet (between 1,000 and 2,000 colonies per plate) and incubated overnight at 37°C (Fig. S2). The colonies were harvested from each plate with a sterile spreader after flooding the plate with 1 mL of PBS, collected into a 50 mL Falcon tube and concentrated by centrifugation at 7,000 × *g* for 5 min. This was repeated for each library. Prior to pooling independent TraDIS libraries, they were separately sequenced (described below) to ensure that an adequate number of

unique insertions was obtained. The independent libraries were pooled, resuspended in BHI containing 2.5 M glycerol and 10 µg/mL of Tet, and 1 mL aliquots were stored at −80°C. The final concentration of cells in the pooled library was determined by dilution and plating onto BHIA containing 10 µg/mL of Tet.

### TraDIS library challenge

The pooled AUS0233ΔtetM transposon mutant library was diluted to an $OD_{600}$ 0.2 in 10 mL of BHI broth containing 10 µg/mL of Tet and incubated at 37°C with shaking until $OD_{600}$ 0.4 (1.5–2 h) (Fig. 2). The culture was diluted to $OD_{600}$ 0.005 in 10 mL BHI broth (50 mL Falcon tube) with or without 4 µg/mL vancomycin (in duplicate). The starting inoculum was $1.0 \times 10^6$ CFU/mL with these cultures incubated at 37°C with shaking until $OD_{600}$ 1.5 (6.5–7.25 h—eight generations: ~$2.5 \times 10^8$ CFU/mL) (Fig. S2). A 2 mL aliquot from each culture was concentrated (7,000 × $g$ for 2 min), washed with 1 mL PBS, and genomic DNA extracted.

### Genomic DNA extraction

The above cell pellet was resuspended in 100 µL of lysis buffer (PBS containing 30 µL of lysozyme [100 mg/mL] and 10 µL of RNaseA [10 mg/mL]), then 100 µL of tissue lysis buffer (Monarch Genomic DNA Purification kit, NEB) was added and vortexed for 10 s. The cells were incubated at 37°C for 1 h. A 10 µL volume of Proteinase K (20 mg/mL) was added, vortexed for 10 s and incubated at 56°C for 30 min at 1,400 rpm. Samples were then processed following the manufacturer's instructions for Monarch Genomic DNA Purification Kit.

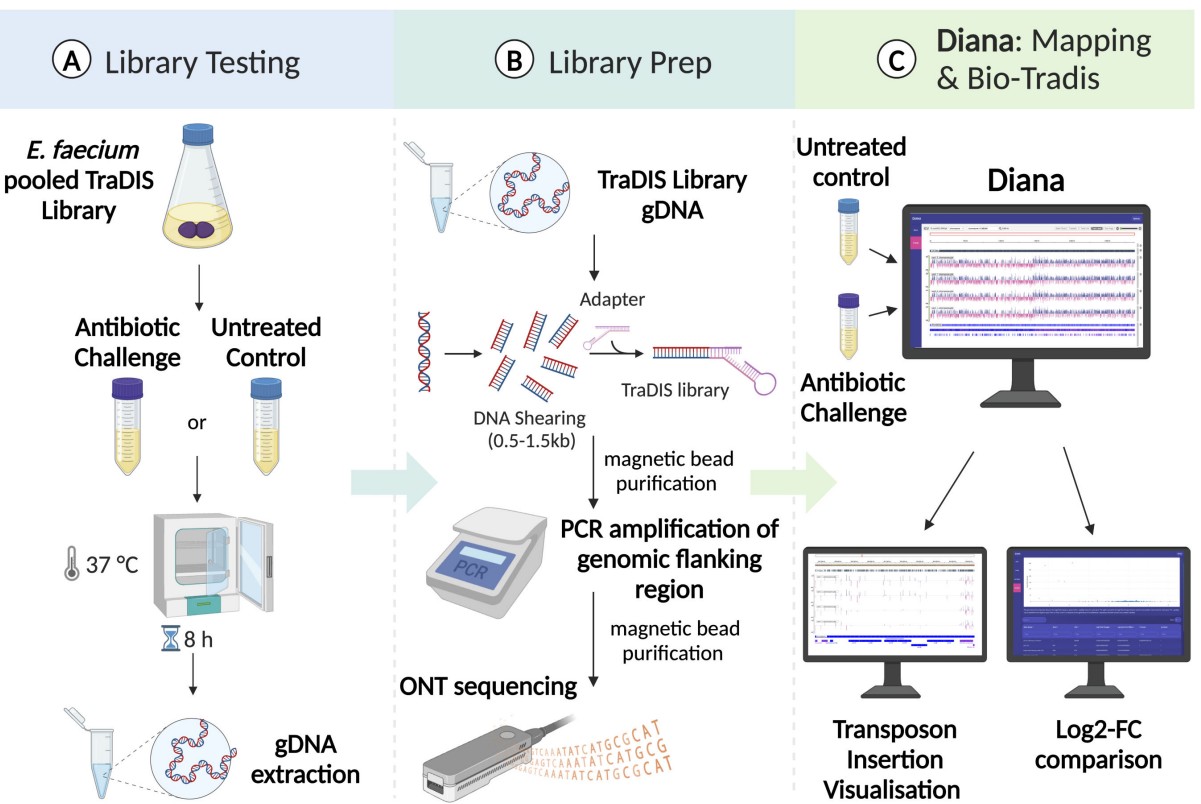

**FIG 2** TraDIS library testing method. (A) Challenging the TraDIS library with antibiotic exposure and extracting genomic DNA. (B) Shearing of genomic DNA to 0.5–1.5 kb, end prep and adaptor ligation. PCR amplification of the transposon/adaptor-flanked genomic DNA and ONT sequencing. TraDIS amplicon reads were trimmed, and genomic reads were mapped to the reference sequence. Transposon insertions are plotted and quantified, shown as the red and blue lines using Diana. The red and blue lines signify strand insertion. The untreated and experimental conditions were examined using the Diana compare function (Bio-Tradis) to find genes impacted under challenge conditions.

## Shearing of genomic DNA

The extracted genomic DNA was diluted to 10 ng/µL in 400 µL milliQ water and aliquoted across two Biorupter tubes (Diagenode). The Biorupter Pico (Diagenode) was run for 12 cycles with 30 s on and 30 s off at 4°C. To confirm adequate shearing (spread between 0.5 and 1.5 kb), 20 µL of sheared genomic DNA was run on a 2% TAE gel at 120 V for 30 min. The two tubes of sheared genomic DNA were combined and concentrated using the Monarch PCR and DNA Clean Up Kit (NEB) and eluted in 15 µL. The eluted sheared genomic DNA was run on a 2% TAE gel for 45 min at 75 V and the region between 0.5 and 1.5 kb was gel extracted and processed using the Monarch DNA Gel Extraction Kit (NEB). This routinely obtained at least 300 ng of DNA per sample, which was qualitatively analyzed on a TapeStation 4200 (Agilent).

### End prep and adaptor ligation for nanopore sequencing

For each TraDIS library sample, 250 ng of sheared DNA was used as input into the NEBNext Ultra II DNA Library Prep Kit for Illumina (NEB). The splinkerette adaptor was prepared by combining 20 µL of IM1080 (50 µM), 20 µL of IM1081 (50 µM), 50 µL water, and 10 µL 10× annealing buffer (100 mM Tris pH 8; 10 mM EDTA; and 500 mM NaCl). The oligonucleotides were annealed by heating to 96°C for 2 min and then reduced by 0.1°C per sec to 22°C. A 2.5 µL aliquot of the splinkerette (instead of the NEBNext adaptor) was ligated to the end-repaired DNA, with the unbound adaptor removed by size selection (0.6 × ratio of Ampure XP beads, Agencourt). The DNA was eluted in 17 µL of elution buffer (10 mM Tris-HCl pH 8.5) and quantified using the Qubit BR dsDNA Kit (Thermofisher).

### PCR amplification of TraDIS amplicon

The following PCR was assembled to amplify the junction region between the transposon insertion and splinkerette: Primers IM1757/IM1758 (200 nM) and 50 ng of the library. A 21-cycle Phusion PCR was conducted: 1 × 98°C to 30 s, 21× (98°C to 10 s, 55°C to 10s, and 72°C to 30s) and a final 40 s at 72°C. A 10 µL aliquot of each amplified library and negative control PCR of AUS0233Δ*tetM* genomic DNA as PCR template was run on a 2% TAE gel for 45 min at 75V. The remaining PCR reaction was purified with AMPure XP beads (as above) and eluted in 17 µL of elution buffer. The DNA was quantified using the Qubit BR dsDNA Kit with a minimum of 200 ng of the TraDIS amplicon required for nanopore sequencing (Fig. 2).

## Nanopore sequencing of TraDIS amplicons

To sequence full-length amplicons, Oxford nanopore technology (ONT) libraries were prepared using the ONT SQK-NBD114-96 Kit, and the resultant libraries were sequenced using R10.4.1 MinION flow cells (FLO-MIN114) on a Linux-based GPU-enabled computer. ONT data were base called using the dna_r10.4.1_e8.2_sup@v4.3.0 model (Dorado v0.7.2) (31).

## TraDIS amplicon sequence processing

The ONT reads obtained from the TraDIS amplicon were trimmed using a *Python* script (extract_tradis_ont_reads.py), which trims the transposon sequence (user specified) and adaptor sequence (user specified) leaving the genomic reads (Fig. 3). The script also removes reads that match to 20 bp of the ITR flanking plasmid sequence (negative fasta), yielding trimmed_reads.fastq. The trimmed_reads.fastq files were mapped to the AUS0233Δ*tetM* genome using *minimap2* (v2.28) (32). The resulting alignment *paf* files (Pairwise mApping Format) were processed by a second *Python* script (create_plot_files.py) to generate plot files. The script maps the transposon amplicon reads to the reference genome, ensuring that the alignment begins at the start of the read to minimise split reads being counted as multiple insertions (--max_gap 5) and that there is sufficient match identity (--min_id 95). Sites that have fewer than two insertions were

excluded from the mapping analysis (--exclude_sites_below two reads). The output of *create_plot_files.py* also provides statistics on the number of TA sites across the genome, assessment of alignment quality, number of insertions in the genome, and the percentage of TA sites with insertions. Plots of the transposon insertion site were visualised using *Diana*, a WebAssembly tool for exploring TraDIS data (see following section; https://diana.cpg.org.au/). The extract_tradis_ont_reads.py and create_plot_files.py scripts are available with documentation on GitHub (https://github.com/rrwick/TraDIS-ONT).

## Web interface to visualize plots and analyze fitness (Diana)

A TraDIS genome-wide exploration tool called "*Diana*" was developed to facilitate visualization and analysis of TraDIS transposon insertions across a reference bacterial genome (https://diana.cpg.org.au/). We are preparing a separate, technical paper on Diana (https://github.com/Wytamma/diana). Here, we present only the essential details with respect to the *E. faecium* application described. User-provided annotated genomes (GenBank or GFF format) and insertion site data (userplot or wiggle format) obtained from the from the scripts described in the section above are processed locally in the browser using WebAssembly versions of bioinformatic modules such as Samtools provided by biowasm (biowasm.com) (33). The results are visualised with the JavaScript implementation of the *Integrative Genomics Viewer* (34). In addition to insertion visualization, the software integrates the functionality for *Bio-Tradis* computational analysis, allowing users to compare two conditions and identify conditionally essential genes. The *Bio-Tradis* R script *tradis_comparisons.R* employs *EdgeR* (https://github.com/OliverVoogd/edgeR) and compares insertion read counts between two conditions. A modified version of this script is run in Diana using a WebAssembly version of the R programming language, WebR (35). The output identifies differences in transposon insertion frequency between the two conditions for each gene, reported as: log count per million (logCPM), log-2 fold-change (FC), with adjusted *P* values (Q-values) for each comparison. Differences were considered significant using these criteria: >5 logCPM, log2-FC of ≥1, ≤−1, <0.001 *q* value (36). Furthermore, the *Bio-Tradis tradis_essentiality.R* script was then used to define essential genes (8, 17, 37).

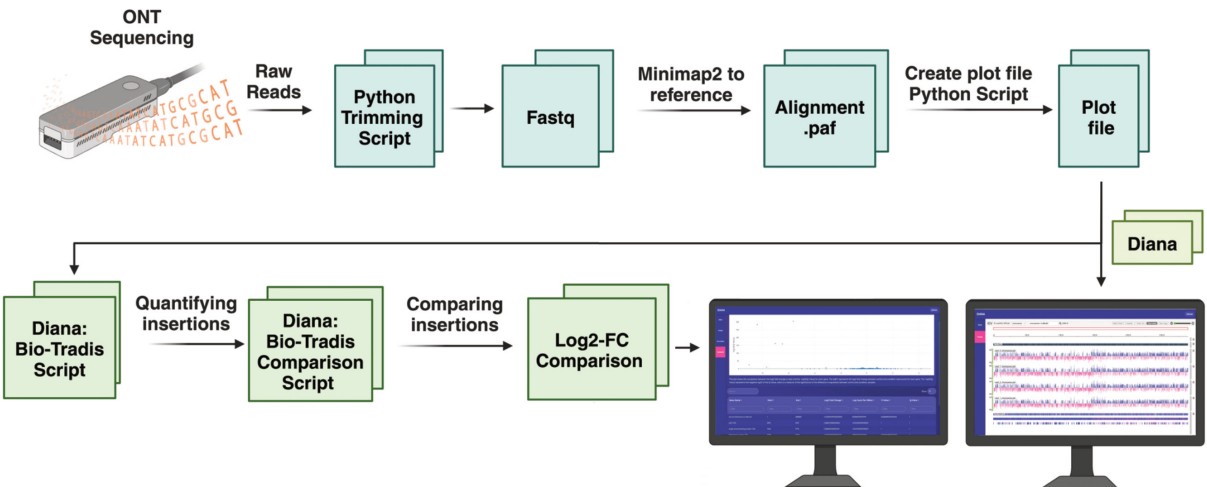

**FIG 3** Flow chart of TraDIS amplicon sequence analysis. TraDIS amplicon reads were trimmed and then put through create_plot_files.py to generate insertion plot files. These plot files are then used in "Diana," which uses a Bio-Tradis R script to compare the insertion read count for each experimental condition. Results were reported as log2-FC, logCPM, with summary statistics for transposon insertion frequency between the experimental conditions. This allows for the interpretation of gene essentiality in different conditions.

## Analysis of published VREfm TraDIS sequence data

To compare our TraDIS library to the previously published Tn-Seq data, we obtained the raw Illumina reads (ENA accession PRJEB19025), which were trimmed using *fastx-trimmer* (v0.0.13) to remove the first 7 bp and the last 30 bp (18, 38). *Bowtie2* (v2.5.4) was used to map the trimmed reads to the *E. faecium* E745 reference to generate a SAM file (39). This SAM file was converted to a *.paf* file using *paftools.js* in *Minimap2* (32). The resulting *.paf* file was used to create a plot file for each TraDIS library using *create_plot_files.py*.

## RESULTS AND DISCUSSION

### The pIMTA(*tetM*) plasmid for TraDIS library construction in *E. faecium*

Despite repeated attempts with the VanB clinical ST796 VREfm isolate AUS0233, we were unable to construct an *E. faecium* transposon library using the previously described plasmid pGPA1 (18), with the main issue related to high plasmid retention. In the present study, we constructed a new transposon delivery plasmid with tetracycline-inducible expression (rather than the nisin induction system used in pGPA1) of the C9 variant of the *himar1* transposase (27). Instead of the MmeI modified inverted terminal repeats (ITRs) present in pGPA1, we used the unmodified ITR sequences. The endogenous P*tetM-tetM* was deleted from VREfm AUS0233, yielding AUS0233Δ*tetM* and allowing the selection of Tet-resistant transposon mutants. Deletion of P*tetM-tetM* had no impact on the growth of the strain in BHI compared to wild type (Fig. S3A). The plasmid pIMTA(*tetM*) transformed efficiently into AUS0233Δ*tetM* at ~1.0 × $10^3$ CFU/μg and we subsequently established a simple protocol to construct high-density transposon libraries in AUS0233Δ*tetM* (see Materials and Methods). The performance was initially assessed through sequencing 10 randomly selected tetracycline-resistant colonies. We could show that each colony had a single, unique transposon insertion with all 10 insertions occurring at different TA sites (Table 2).

### Optimization of transposon mutant library construction

For TraDIS library construction, we initially optimized the level (aTc; 0–500 ng/mL) and duration (1 and 24 h) of transposase induction. We observed that induction with 500 ng/mL of aTc for 24 h was required for maximal transposition, with a reduction in transposition observed when aTc or induction time was decreased. To further improve plasmid loss, we compared the impact of temperature (37°C vs 42°C) combined with the above optimized induction parameters. Non-selective outgrowth of the library at 37°C post-induction with 500 ng/mL of aTc yielded ~95% plasmid loss, with this increasing to >98% plasmid loss at 42°C. We then evaluated the impact of cell density from plate sweeps (~1,000 vs ~10,000 CFU per plate across 40 plates) on the number of unique transposon insertions detected. However, sequence analysis showed only a slight increase in the number of unique insertion sites between the ~4 × $10^4$ (9,596 unique insertions) and ~4 × $10^5$ (13,514 unique insertions) pooled colonies (Table S1: Sheet 5), which could be due to the saturation of unique insertions present in a single induction experiment. Therefore, pooling of independent library inductions is important in the construction of large transposon mutant libraries. Here, we pooled a total of six independent inductions to generate our final TraDIS library in AUS0233Δ*tetM,* which contained a total of 48,458 unique insertions, with 37,156 mapping to the chromosome and 11,302 to plasmids within AUS233Δ*tetM* (Fig. 4A; Table S1: Sheet 1). Taking advantage of the recent improvements in the quality of ONT sequencing with the R10.4.1 chemistry and base calling (40, 41), we were able to apply nanopore amplicon sequencing to reliably identify the genomic/transposon junctions. We tested two primer pairs for the junction amplification with outward-facing primers from either side of the transposon (IM1756 or IM1757) with the adapter primer IM1758. No differences in amplification or sequencing bias were observed, with the primer pair IM1757/IM1758. ONT has been used before in transposon insertion site sequencing to overcome limitations in determining transposon insertion sites in large repeat regions in the genome (42). In

**TABLE 2** Single colony transposon insertion locations across the AUS0233Δ*tetM* genome[b]

| Colony no. | Insertion location | Strand | Insertion position within gene (bp) | Insertion co-ordinates[a] |
|---|---|---|---|---|
| 1 | Intergenic region (plasmid CP176376) | + | - | 103,875 bp |
| 2 | Bacterial Ig-like domain-containing protein (chromosome) | + | 1,294/2,061 | 723,984 bp |
| 3 | RpiR family transcriptional regulator (chromosome) | + | 254/849 | 1,061,595 bp |
| 4 | Intergenic region (chromosome) | + | - | 187,574 bp |
| 5 | ATP-binding protein (chromosome) | – | 174/1,449 | 1,819,804 bp |
| 6 | PRD domain-containing protein (chromosome) | + | 438/873 | 2,717,223 bp |
| 7 | PAP2 family protein (chromosome) | + | 495/657 | 2,597,497 bp |
| 8 | Alkaline phosphatase family protein (plasmid CP176376) | – | 33/825 | 125,402 bp |
| 9 | PTS system IID component (chromosome) | + | 143/831 | 279,110 bp |
| 10 | Hypothetical protein (plasmid CP176378) | + | 1,194/2,055 | 17,680 bp |

[a]Insertion in AUS0233Δ*tetM* genome.
[b]'+' and '-' denote that the insertion is on the positive or negative strand.

the application of ONT sequencing here, we could routinely sequence eight TraDIS libraries together on a MinION, generating an average of 20 million reads for each run (range of 1–3 million reads per sample). Advantages of ONT sequencing for TraDIS compared to Illumina sequencing include not needing to bypass factory settings on the MiSeq instrument to permit dark cycling, while also offering a simpler process for library preparation and sequencing (43–45).

## Comparison of the AUS0233ΔtetM and E745 transposon libraries

Using our insertion site mapping approach, we next undertook a comparison of our data set with the previously published VREfm Illumina Tn-Seq library constructed using pGPA1 (18). This analysis revealed a total of 36,183 unique VREfm E745 genome insertions, comprising 30,208 unique chromosomal insertions (Fig. 4A; Table S1: Sheet 2). The frequency of plasmid loss for our library and the previous pGPA1 library was assessed by read mapping to the ITR adjacent plasmid sequence. This comparison revealed 94.46% plasmid loss for pGPA1 and 98.22% plasmid loss for pIMTA(*tetM*). This latter observation is consistent with the >98% plasmid loss estimated from our colony patching of pIMTA(*tetM*) (see above). We also noted a leading-lagging chromosome strand insertion bias present in our library that was less obvious in the E745 library (46, 47) (Fig. 4A). This absence of strand bias and the presence of insertions in known essential genes for low-GC Gram-positive bacteria, such as *dnaA*, *dnaB*, *dnaN*, *gyrB,* and *gyrA*, in the E745 library might be explained by differences in transposase induction strategies and subsequent sub-culturing methods (48, 49).

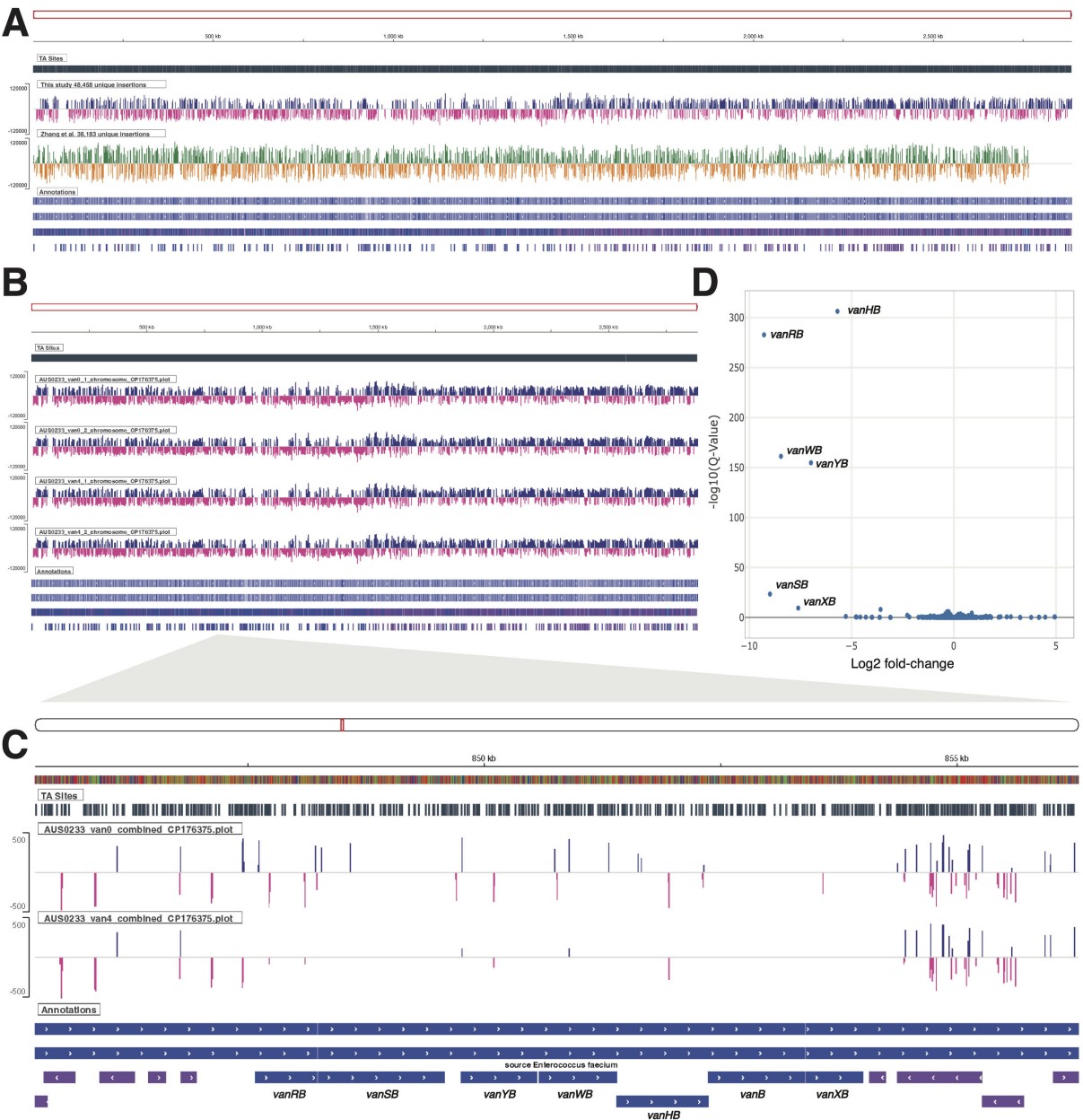

**FIG 4** Performance of pIMTA(*tetM*) AUS0233Δ*tetM* TraDIS library. (A) Diana plot tracks showing unique transposon insertions for pIMTA(*tetM*) transposon library across the AUS0233Δ*tetM* chromosome (this study) compared with a previously published VREfm Tn-Seq library (18). (B) Diana AUS0233Δ*tetM* chromosome plot track for the pIMTA(*tetM*) transposon library after growth in the presence of vancomycin at 0 and 4 µg/mL. (C) A zoomed-in Diana plot track of combined libraries showing transposon insertions across the AUS0233Δ*tetM vanB* locus. (D) Diana volcano plot showing log-2 based fold change and adjusted *P* values (*Q* values) for transposon insertion frequency differences between vancomycin-exposed and control TraDIS libraries.

## Identifying genes essential for vancomycin resistance

To test the performance of our AUS0233Δ*tetM* transposon library, we performed TraDIS after growth of the library in the absence or presence of vancomycin at 4 µg/mL, the clinical breakpoint for defining vancomycin-susceptible *E. faecium* and VREfm (50). This concentration of vancomycin did not impact the growth of the transposon library compared to untreated control (Fig. S3B). Strain AUS0233Δ*tetM* contains the chromosomal *vanB* operon which consists of seven genes (*vanR, vanS, vanY, vanW, vanH, vanB,* and *vanX*) (51). These genes create a modified peptidoglycan precursor pentadipepsipeptide that incorporates a terminal D-Ala–D-Lac instead of D-Ala–D-Ala, the latter moiety being

the target of vancomycin. The TraDIS sequencing data were processed using the pipeline developed (Fig. 3) and analyzed using Bio-Tradis in *Diana* to identify the key genes involved in vancomycin resistance (Table S2). From this analysis, we confirmed the role of the *vanB* operon in vancomycin resistance (51, 52), with four of the seven genes (*vanYB*, *vanRB*, *vanWB,* and *vanHB*) having a significantly higher frequency of transposon insertions in control versus vancomycin-treated library (Table 3; Fig. 4C and D) (53–55). This result also confirms the functionality of the transposon library. Of the remaining three genes of the *vanB* operon in AUS0233, *vanSB* and *vanXB* contained no transposon insertions under vancomycin exposure and only two or one insertion in the control, respectively (Fig. 4C). The genes *vanSB* and *vanXB* met the established significance thresholds except for logCPM, which was low (3.05 and 1.77 logCPM, respectively) (Table 3; Fig. 4D). Interestingly, we observed no transposon insertions in *vanB* in the original library (Fig. 4C), and we are currently following up on the apparent synthetic lethality of *vanB* loss in the presence of a functional *vanB* operon. In *vanB E. faecium*, *vanHBX* are considered 'core' vancomycin resistance genes because they encode the dehydrogenase (*vanH*) responsible for converting pyruvate to lactate, the ligase (*vanB*) that synthesises the D-Ala–D-Lac peptide and the dipeptidase (*vanX*), which hydrolyzes the pre-existing D-Ala–D-Ala (56). Similarly, the two-component regulatory system encoded by *vanRS*, with a sensor histidine kinase (VanS) that detects vancomycin and its response regulator VanR that upregulates transcription of the operon, is also considered core (57, 58). Nothing is known of the function of *vanYB* and *vanWB* in *E. faecium*. However, in *vanA E. faecalis* and *E. faecium*, *vanW* is absent and *vanY* is considered an accessory gene, as it is not required to confer vancomycin resistance (51, 58–60). However, here we have shown that *vanYB* and *vanWB* likely play a more significant role in the response to vancomycin in the *vanB* operon compared to the *vanA* operon, as both genes were significantly unrepresented with transposon insertions after exposure to the antibiotic compared with control (Fig. 4D). This result highlights the utility of the TraDIS approach developed to identify conditionally essential genes and further probe gene function. We did not identify any significant hits from the screen outside of the VanB operon.

## Conclusions

In this study, we successfully generated a functional TraDIS library platform optimized for *E. faecium* using the newly designed pIMTA(*tetM*) plasmid, overcoming the challenges encountered with a previous transposon mutagenesis system developed for VREfm (18). Our refined TraDIS library method enabled the generation of unique transposon mutants with >98% plasmid loss, ensuring efficient transposase induction and minimal sequencing contamination with pIMTA(*tetM*). The use of nanopore amplicon sequencing provided a convenient and cost-efficient alternative to Illumina sequencing, enabling the identification of unique transposon insertion sites across the genome. Using this transposon mutant library, we confirmed the VanB operon to be essential for vancomycin resistance in VanB *E. faecium*, validating the power of TraDIS as a tool to identify key genes involved in survival under challenge conditions. The success of this approach underscores its potential as a versatile tool for functional genomics. The presence of

**TABLE 3** Bio-Tradis analysis AUS0233Δ*tetM* genes involved survival after vancomycin challenge[b]

| Gene | Function[a] | Log2-FC | LogCPM | adj-*P* | Comments |
|------|-------------|---------|--------|---------|----------|
| *vanRB* | DNA-binding response regulator | −9.28 | 6.48 | 1.54e−283 | - |
| *vanWB* | Unknown | −8.45 | 5.67 | 5.16e−162 | - |
| *vanYB* | D-alanyl–D-alanine carboxypeptidase | −6.98 | 5.69 | 1.45e−155 | - |
| *vanHB* | D-specific alpha-keto acid dehydrogenase | −5.69 | 6.81 | 4.08e−307 | - |
| *vanSB* | Sensor histidine kinase | −8.99 | 3.05 | 4.03e−24 | <5 logCPM |
| *vanXB* | D-alanyl–D-alanine dipeptidase | −7.61 | 1.77 | 3.98e−10 | <5 logCPM |
| *vanB* | D-alanine–D-lactate ligase | - | - | - | <5 logCPM, log2-FC of ≤1, ≥−1, >0.001 *q* value |

[a]Reference 51.
[b]'-' denotes a negative fold change.

the complete, broadly permissive, pWV01 temperature-sensitive replicon should allow the application of the pIMTA(*tetM*) to other low G + C Gram-positive bacteria where ATc induction can be applied (21, 26). Building on these findings, this tool could be used in future studies to investigate the genetic determinants of other prominent phenotypes in *E. faecium* such as acid or alcohol tolerance (61).

## AUTHOR AFFILIATIONS

[1]Department of Microbiology and Immunology, The University of Melbourne at The Peter Doherty Institute for Infection and Immunity, Melbourne, Victoria, Australia

[2]Centre for Pathogen Genomics, Doherty Institute, The University of Melbourne, Melbourne, Victoria, Australia

[3]Department of Microbes, Infection and Microbiomes, School of Infection, Inflammation and Immunology, College of Medicine and Health, University of Birmingham, Birmingham, United Kingdom

[4]Microbiological Diagnostic Unit Public Health Laboratory, The University of Melbourne at The Peter Doherty Institute for Infection and Immunity, Melbourne, Australia

## AUTHOR ORCIDs

Timothy P. Stinear  http://orcid.org/0000-0003-0150-123X
Andrew H. Buultjens  http://orcid.org/0000-0002-5984-1328
Ian R. Monk  http://orcid.org/0000-0001-6982-8074

## FUNDING

| Funder | Grant(s) | Author(s) |
| --- | --- | --- |
| National Health and Medical Research Council | GNT1194325 | Timothy P. Stinear |
| National Health and Medical Research Council | GNT1185213 | Glen P. Carter |
| Biotechnology and Biological Sciences Research Council | APP21400 | Willem van Schaik |

## AUTHOR CONTRIBUTIONS

Alexandra L. Krause, Data curation, Formal analysis, Investigation, Writing – original draft, Writing – review and editing | Wytamma Wirth, Data curation, Investigation, Methodology, Resources, Software, Writing – original draft | Adrianna M. Turner, Investigation, Methodology, Validation, Writing – review and editing | Louise Judd, Investigation, Methodology | Willem van Schaik, Resources, Writing – review and editing | Benjamin P. Howden, Supervision, Writing – review and editing | Glen P. Carter, Supervision, Writing – review and editing | Torsten Seemann, Resources, Software, Supervision | Ryan Wick, Data curation, Formal analysis, Investigation, Methodology, Resources, Software, Validation, Writing – review and editing | Timothy P. Stinear, Conceptualization, Funding acquisition, Investigation, Methodology, Project administration, Resources, Software, Supervision, Writing – review and editing | Andrew H. Buultjens, Formal analysis, Investigation, Methodology, Supervision, Validation | Ian R. Monk, Conceptualization, Formal analysis, Investigation, Methodology, Supervision, Writing – review and editing.

## DATA AVAILABILITY

The sequence of pIMTA(*tetM*) is available from NCBI (GenBank accession no. PV061376). The AUS0233Δ*tetM* genome sequence deposited under the accession number PRJNA1196086. The transposon library reads are available from NCBI (GenBank accession no. PRJNA1196996).

## ADDITIONAL FILES

The following material is available online.

## Supplemental Material

**Supplemental figures (Spectrum00628-25-s0001.docx).** Fig. S1 to S3.
**Table S1 (Spectrum00628-25-s0002.xlsx).** TraDIS sequencing stats.
**Table S2 (Spectrum00628-25-s0003.xlsx).** Bio-Tradis analysis of the vancomycin exposed library.

## Open Peer Review

**PEER REVIEW HISTORY (review-history.pdf).** An accounting of the reviewer comments and feedback.

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
