## [Reviewer comments · Microbiology Spectrum]

Microbiology Spectrum

Transposon-directed insertion-site sequencing (TraDIS) analysis of *Enterococcus faecium* using nanopore sequencing and a WebAssembly analysis platform

Alexandra Krause, Wytamma Wirth, Adrianna Turner, Louise Judd, Lucy Li, Willem Van Schaik, Benjamin Howden, Glen Carter, Torsten Seemann, Ryan Wick, Timothy Stinear, Andrew Buultjens, and Ian Monk

Corresponding Author(s): Ian Monk, The University of Melbourne

Review Timeline:

Submission Date:	March 3, 2025
Editorial Decision:	April 21, 2025
Revision Received:	April 30, 2025
Accepted:	May 11, 2025

Editor: John Atack

Reviewer(s): Disclosure of reviewer identity is with reference to reviewer comments included in decision letter(s). The following individuals involved in review of your submission have agreed to reveal their identity: emma rachel Holden (Reviewer #2)

Transaction Report:

DOI: <https://doi.org/10.1128/spectrum.00628-25>

Re: Spectrum00628-25 (**Transposon-directed insertion-site sequencing (TraDIS) analysis of *Enterococcus faecium* using nanopore sequencing and a WebAssembly analysis platform**)

Dear Dr. Ian Robertson Monk:

Thank you for the privilege of reviewing your work. Below you will find my comments, instructions from the Spectrum editorial office, and the reviewer comments.

Revision Guidelines

Sincerely,
John Attack
Editor
Microbiology Spectrum

Reviewer #1 (Comments for the Author):

The manuscript by Krause et. al. describes the development and implementation of a new TraDIS platform for *E. faecium*. A himar1 TraDIS approach has been previously deployed in other *E. faecium* strains by other groups, however the authors report failure with this method using their clinical isolates, leading to the need to develop a new platform. The new approach uses a hyperactive himar1 variant that is anhydrotetracycline inducible, combined with Nanopore amplicon sequencing to identify the

transposon insertion sites, rather than Illumina as is traditionally used in TraDIS. The new platform was validated using a vancomycin challenge to detect genes required for vancomycin resistance.

The body of work is comprehensive. The authors have undertaken significant experimental optimisation and validation. The manuscript is well-written, and it is accompanied by appropriate figures to explain the methodology and bioinformatic pipelines deployed. The development of a new TraDIS platform addresses the failure of the existing *E. faecium* TraDIS approach with some clinical strains (as yet unexplained), and adds utility to the field. I have no further comments or corrections.

Reviewer #2 (Comments for the Author):

In this manuscript, the authors describe construction of a novel TraDIS library in *Enterococcus faecium*, validate library efficacy through Vancomycin stress testing and introduce a data analysis online platform 'Diana'. Overall, this work has been well executed and presented and is of a high quality. My only comment is that this work would present a more complete story and could likely reach a higher impact journal and if combined with the manuscript in preparation on the Diana data analysis platform. If time and resources permit this may be worth consideration.

Reviewer #3 (Comments for the Author):

In this manuscript Krause et al. present an approach for generating transposon insertions libraries in the relatively genetically intractable organism *E. faecium*. Despite tools like this existing for *E. faecium*, their utility can vary among *E. faecium* strains, especially contemporary clinical isolates. The authors develop a new plasmid, pIMTA(tetM), that is used for transposon-directed insertion site sequencing (TraDIS), that is then coupled to a computational pipeline for identifying Tn insertions using long read sequencing. Overall, this is a very well written manuscript with clear methodology that will be immensely beneficial for the *Enterococcus* field. I have a few comments for the authors moving forward.

1. It is intriguing that *vanW* and *vanY* are enriched for Tn insertions within the *vanB* locus. The authors make the differentiation between *vanB* and *vanA* in that *vanW* and *vanY* are absent in *vanA* enterococci. The authors also indicate that nothing is known about these genes and their function. I would suggest that the authors bolster this claim in their narrative, as it appears their method has utility in identifying genes of critical function in enterococci that are understudied.
2. Can the authors comment on the other genes that acquire Tn insertion site enrichment outside of *vanB* during vancomycin exposure. Does this reveal anything unique or unexpected in relation to vancomycin resistance in *E. faecium*?
3. As only one clinical strain is tested here, it would be helpful to know how broadly useful this plasmid system is among *E. faecium* strains, and even in *E. faecalis*. For the latter, such tools exist, but do suffer from similar issues as raised about the pGPA1 system. Perhaps a few additional strains of interest could be tested and insertions sites verified, similar to what is shown in table 1.
4. Can the authors comment on how one might implement an Illumina based strategy for the analysis of insertion sites, were someone not to have access to ONT and need to use an Illumina based approach. Can this be implemented into the current bioinformatic tools for instance?

Response to reviewer comments

We have carefully considered each of the issues raised by the reviewers. Please find our point-by-point responses below. Page and line numbers refer to the marked-up, revised manuscript. Additional text is highlighted in yellow.

Reviewer #1 (Comments for the Author):

The manuscript by Krause et. al. describes the development and implementation of a new TraDIS platform for *E. faecium*. A himar1 TraDIS approach has been previously deployed in other *E. faecium* strains by other groups, however the authors report failure with this method using their clinical isolates, leading to the need to develop a new platform. The new approach uses a hyperactive himar1 variant that is anhydrotetracycline inducible, combined with Nanopore amplicon sequencing to identify the transposon insertion sites, rather than Illumina as is traditionally used in TraDIS. The new platform was validated using a vancomycin challenge to detect genes required for vancomycin resistance.

The body of work is comprehensive. The authors have undertaken significant experimental optimisation and validation. The manuscript is well-written, and it is accompanied by appropriate figures to explain the methodology and bioinformatic pipelines deployed. The development of a new TraDIS platform addresses the failure of the existing *E. faecium* TraDIS approach with some clinical strains (as yet unexplained), and adds utility to the field. I have no further comments or corrections.

RESPONSE: None required.

Reviewer #2 (Comments for the Author):

In this manuscript, the authors describe construction of a novel TraDIS library in *Enterococcus faecium*, validate library efficacy through Vancomycin stress testing and introduce a data analysis online platform 'Diana'. Overall, this work has been well executed and presented and is of a high quality. My only comment is that this work would present a more complete story and could likely reach a higher impact journal and if combined with the manuscript in preparation on the Diana data analysis platform. If time and resources permit this may be worth consideration.

RESPONSE: We considered including more detail on building the TraDIS analysis platform Diana. However, the technical details of using WebAssembly to integrate the various data analysis modules would have expanded the manuscript substantially. In addition, this information will largely be of interest to a different audience to those that will use our TraDIS approach. Note that we do provide information in the methods on how Diana was engineered, and we provide links to the underlying code and software used. (Page 13, line 244).

Reviewer #3 (Comments for the Author):

In this manuscript Krause et al. present an approach for generating transposon insertions libraries in the relatively genetically intractable organism *E. faecium*. Despite tools like this existing for *E. faecium*, their utility can vary among *E. faecium* strains, especially contemporary clinical isolates. The authors develop a new plasmid, pIMTA(tetM), that is used for transposon-directed insertion site sequencing (TraDIS), that is then coupled to a computational pipeline for identifying Tn insertions using long read sequencing. Overall, this is a very well written manuscript with clear methodology that will be immensely beneficial for the *Enterococcus* field. I have a few comments for the authors moving forward.

1. It is intriguing that vanW and vanY are enriched for Tn insertions within the vanB locus. The authors make the differentiation between vanB and vanA in that vanW and vanY are absent in vanA enterococci. The authors also indicate that nothing is known about these genes and their function. I would suggest that the authors bolster this claim in their narrative, as it appears their method has utility in identifying genes of critical function in enterococci that are understudied.

RESPONSE: This is an important point. We have added the following text to the manuscript in yellow:

ABSTRACT: As expected, we could confirm the importance of the *vanB* operon for VREfm vancomycin resistance, however we also identified an essential role for both *vanWB* and *vanYB*, each previously designated as protein of unknown function and accessory for resistance, respectively. (Page 1, line 30-32).

RESULTS: This result highlights the utility of the TraDIS approach developed to identify conditionally essential genes and further probe gene function. (Page 19, line 371-373).

2. Can the authors comment on the other genes that acquire Tn insertion site enrichment outside of vanB during vancomycin exposure. Does this reveal anything unique or unexpected in relation to vancomycin resistance in *E. faecium*?

RESPONSE: *With the conditions of the screen employed, we did not identify any genes outside of the VanB operon as playing a role in vancomycin resistance. This could be due to the breakpoint MIC chosen, with a lower concentration permitting the identification of accessory genes. We have added the following sentence at the end of the results section.*

We did not identify any significant hits from the screen outside of the VanB operon. (Page 19, line 373).

3. As only one clinical strain is tested here, it would be helpful to know how broadly useful this plasmid system is among *E. faecium* strains, and even in *E. faecalis*. For the latter, such tools exist, but do suffer from similar issues as raised about the pGPA1 system. Perhaps a few additional strains of interest could be tested and insertions sites verified, similar to what is shown in table 1.

RESPONSE: *We have used the pIMTA(tetM) plasmid to generate a large library in a second *E. faecium* strain (VanA – ST1421), but this is for a different study. The entire pWV01 temperature sensitive replicon region used in pIMTA(tetM) is known to be functional in a number of low G+C Gram positive bacteria (such as *Enterococcus faecalis*, *Listeria monocytogenes*, *Staphylococcus aureus*, *Streptococcus pyogenes*). We have added the following sentence to the discussion to highlight this point.*

The use of the complete, broadly permissive pWV01 temperature sensitive replicon should allow the application of the pIMTA(tetM) to other low G+C Gram-positive bacteria where ATc induction can be applied (21, 60). (Page 20, line 391-394).

4. Can the authors comment on how one might implement an Illumina based strategy for the analysis of insertion sites, were someone not to have access to ONT and need to use an Illumina based approach. Can this be implemented into the current bioinformatic tools for instance?

RESPONSE: *For the mapping of Illumina reads, the Diana platform can be used as we have shown in Figure 4A with the Illumina Tn-Seq reads from the Zhang et al (2017) study. In the materials and methods, we highlight a different trimming and mapping approach which needs to be applied. We have now highlighted that these were Illumina reads in the results text.*

Using our insertion site mapping approach, we next undertook a comparison of our dataset with the previously published VREfm **Illumina** Tn-Seq library constructed using pGPA1 (Zhang et al, 2017). **(Page 17, line 326).**

Re: Spectrum00628-25R1 (**Transposon-directed insertion-site sequencing (TraDIS) analysis of *Enterococcus faecium* using nanopore sequencing and a WebAssembly analysis platform**)

Dear Dr. Monk:

Your manuscript has been accepted, and I am forwarding it to the ASM production staff for publication. Your paper will first be checked to make sure all elements meet the technical requirements. ASM staff will contact you if anything needs to be revised before copyediting and production can begin. Otherwise, you will be notified when your proofs are ready to be viewed.

Sincerely,
John Attack
Editor
Microbiology Spectrum